# Humanistic Nursing Care for Patients in Low-Resourced Clinical Settings from Students’ Perspectives: A Participatory Qualitative Study

**DOI:** 10.3390/ijerph191912656

**Published:** 2022-10-03

**Authors:** Yanlin Zhu, Gan Liu, Yuqiu Shen, Junqiao Wang, Minmin Lu, Jing Wang

**Affiliations:** School of Nursing, Fudan University, Shanghai 200437, China

**Keywords:** humanistic care, patients, low-resourced environment, nursing students, clinical practicum, participatory qualitative study

## Abstract

Introduction. It is of utmost importance to understand how we can better prepare students to value humanistic spirits and provide humanistic care, a core element of quality care for patients/family characterized by empathy and holistic care, from school to clinical settings (practicum) in low-resourced healthcare environments with seriously low nursing staffing levels. The current study explored participants’ experiences of learning and delivering humanistic care for patients. Methodology. This is a participatory qualitative study. Eligible participants are undergraduate students who completed all the core curricula and are ready to start their one-year clinical practicum/internships. A total of 120 eligible undergraduate students were included in the study. Participants were encouraged to share their own thoughts, questions, and perspectives on learning and providing humanistic care in clinical settings during their one-year practicum from 2020 to 2021. Results. Three major themes emerged: 1. From Textbook to Providing Humanistic Care in low-resourced work contexts; 2. Ethical Considerations; and 3. Implications for Humanistic Nursing Care Education. Discussion. Systematic reforms are needed to make clinical settings more humanistic care-friendly for nurses and nursing students. It is significant to help students internalize the essence of humanistic care in low-resource settings.

## 1. Introduction

Nursing is regarded as a science, profession, and art that emphasizes the nature of caring and has humanistic attributes. Humanistic nursing care is an interaction between nurses and patients/families as a response to the caring situation and is characterized by empathy, respect for human dignity, autonomy of patients, and holistic care. Compared to regular nursing care, humanistic care shifts from a task-oriented care model to a person-centered or relationship-centered model [1,2]. Humanistic care has been considered a core element of nursing education in high-resourced regions and countries around the world [3,4,5,6,7]. Ideally, humanistic nursing care can improve the quality and safety of care for patients/families and promote work satisfaction and well-being for nurses [5,6]. However, there is no fixed method or process to provide humanistic nursing care in clinical settings because humanistic care approaches and behaviors are highly context-specific and should embed a holistic and person-centered perspective to foster humanistic spirits into daily practice [1,8].

This challenge also applies to the nursing profession and education in China. Despite the emphasis of humanistic nursing care, it remains unclear how to sustainably provide high quality humanistic care for patients/family, particularly patients, in the current clinical context. China’s Ministry of Health and the Nursing Association have highlighted the importance of providing humanistic care in clinical settings. The “high-quality nursing service” was launched in 2010 as a nationwide nursing care project to facilitate a humanistic spirit in nursing care [9,10]. The program has raised nursing professionals’ awareness of humanistic care. In order to prepare students from Bachelor of Science in Nursing (BSN) programs to provide humanistic care for patients in clinical settings, top nursing schools in China adapted their pedagogy and core curricula to include more humanistic spirits. For example, concept-based teaching (CBT) was adapted toward a more student-centered pedagogy. The humanistic care model was frequently mentioned. However, it is abstract and not incorporated in the key learning content or grounded in clinical teaching [9,10,11]. Although the nursing profession and higher education have developed rapidly in the past 40 years in China, there is still an acute shortage within the nursing workforce. By the end of 2019, the number of nurses per 1,000 population in China was 3.1, compared to 11.8 in Japan and 11.7 in the US [12]. Registered nurses are still facing challenges implementing humanistic care in their services because of inadequate staffing, a medically-dominated care culture, and the high intensity of work in general hospitals.

It is of utmost importance to understand students’ perspectives so that we can develop a more applied student-centered nursing education program that can better prepare our students to value humanistic spirits and provide humanistic care in their work. The current body of literature focused on faculty members’ experiences of developing or adapting courses or teaching methods to facilitate humanistic care education and the effects of the education on students’ competencies such as “soft skills”, empathy, communicative skills, and so forth [1,4,10,11,13]. However, there is a paucity of literature exploring students’ experiences and perceptions of learning and applying humanistic care from school to clinical settings (practicum) in a country with the largest older population and lower health care resources. The current study will be one of the first studies to adopt a participatory approach in the qualitative study process, including design, data collection, analysis, and interpretation to guarantee the direct engagement of students so that they can actively share their experiences of learning and delivering humanistic care, and their critical reflections on humanistic care education in the undergraduate programs. The aim of the study is to explore nursing students’ experiences of learning and delivering humanistic care during their clinical practicum in a low-resourced context and their critical reflections on humanistic care education. The study will have implications for humanistic care education in other low and middle-income countries.

## 2. Methods

### 2.1. Design

We adopted a participatory approach in the qualitative study process, including design, data collection, analysis, and interpretation [14,15]. The participatory approach is a qualitative research methodology option that values genuine and meaningful participation and direct engagement of local priorities and perspectives. It engages participants who are not necessarily trained in research but belong to or represent those who are the focus of the research through partnerships or shared leadership between researchers and stakeholders or end-users. Currently, participatory qualitative study designs and strategies still vary, but the common value is to conduct research “with” traditionally the “subjects” of the research rather than “on” them [14]. This participatory approach supports us in designing research that is well informed by real-world contexts and gaining real-world knowledge and experiences through mutually reinforcing partnerships with the stakeholders. In the design stage of the study, we invited nursing educators from undergraduate programs and clinical settings, fourth-year students who just completed their one-year practicum, and third-year students who are about to start their practicum (participants of the study) to design the interview guide, reflective journal format, and data collection process through individual interviews, focus groups, and written feedback. We identified the fundamental needs of the research collaboratively through shared decision-making and mutual learning. Interview methods followed the consolidated criteria for the qualitative research report [16].

### 2.2. Participant Selection

Eligible participants are undergraduate students who completed all the core curriculas and are ready to start their one-year clinical practicum/internships: that is, third-year students in a four-year BSN program (*N* = 120, with one student who dropped out because of personal reasons). We included all the third-year undergraduate nursing students who started their clinical practicum in 2020 from a top School of Nursing in China that emphasizes humanistic care in its program descriptions (see Table 1). Data collection was carried out from 2020 to 2021. Some participants were involved in the designing of the study and all the participants were informed of the study purposes and methods of data collection (including observation fieldnotes/memos and reflective journals focusing on humanistic care) before they started their clinical practicum through interactive sessions with the researchers.

### 2.3. Data Collection

Participating students (*n* = 120) were divided into 10 subgroups in the clinical practicum. Each subgroup has a group leader who would be the key point of contact and responsible for collecting field notes/memos and journals every three months until the end of the one-year practicum. Participants were encouraged to keep journals and memos from their own learning and practical experiences and share their own thoughts, questions, and perspectives on learning and providing humanistic care for patients in clinical settings based on the participatory approach. Interview guides were developed based on an initial analysis of the journals and memos through the participatory approach. Nursing educators and students were involved in making shared decisions on the interview guide focus points and questions. We conducted individual interviews with subgroup leaders and other participants and ten focus group interviews with all the participants in their fourth year when they completed their practicum. Focus groups were co-facilitated by some of the participants with interviewers.

The interviewers explored the perspectives of participants regarding their experiences of learning about and delivering humanistic care during their practicum and their critical reflections on humanistic care education in the undergraduate program. The interviewers tailored specific questions and prompts based on the responses from the participants and other emerging information. The interviewers used audio recording to collect the data and did make memos during and after the interview. The interview duration was between 60 and 90 min. Data collection was continued until thematic saturation was achieved based on the inductive approach. Specifically, interviews did not identify new codes for participants’ perceptions and thoughts on the focused topics [17]. A professional transcriptionist transcribed the interviews. De-identified transcripts were uploaded into Dedoose 2018 (http://www.dedoose.com/ (accessed on 1 February 2021)) for analysis.

### 2.4. Data Analysis

Conventional content analyses were applied [18]. This methodological approach was used because it is suitable for learning about and interpreting experiences and perspectives of individuals. The reoccurring codes and subthemes were developed with no preconceived notions. We adopted a participatory qualitative data analysis approach and involved stakeholders (nursing faculty, clinical instructors, and students) in the analysis process so that data were interpreted through the perspective of researchers and community members to produce concrete findings for real-world practice. We used a two-cycle coding approach [19]. The team created and defined initial codes emerging from the data. In the first cycle of coding, we expanded the codes and used in vivo and descriptive coding to identify key points and annotations to keep track of the rationale for coding. Notes were made to clarify initial coding decisions. Any disagreements were resolved in the larger team. After the first cycle of coding, group-level analysis was conducted. Patterns and differences were identified among participants. All of the key points were listed, codes were sorted into sub-categories and then combined into a smaller number of categories. The authors referred to participants’ original quotes wherever more context was needed. Patterns, themes, and relationships between categories were noted in the analysis to develop ideas about themes and interpretation. Discussions among members of the team continued until consensus was reached regarding the themes and interpretations.

### 2.5. Rigor

Multiple strategies were used to establish qualitative rigor [20]. For credibility (confidence in the truth of the findings), we used multiple data sources, including students’ journals of participatory observations and reflections and experiences shared in the interviews. We adopted a qualitative participatory approach so that students had prolonged engagement and persistent observation that could provide scope and depth to the study. We held frequent meetings with the coding team to discuss the use of codes, the coding, and the refined definitions needed to further ensure confirmability (a degree of neutrality). We also engaged other non-researcher team members so that they could review and challenge our interpretations. Using features in Dedoose, we compared our coding patterns to ensure that the interpretations and hypotheses made from the analysis were sound. For transferability (applicability), students were encouraged to keep a detailed account of field experiences so that the researcher could understand the context more explicitly. For dependability (findings are consistent and could be repeated), we kept summaries and memos of all the discussions among members of the coding team to keep track of the interpretation. The refinement of the codes and themes were discussed by the whole team, which facilitated joint decision-making about the coding schemes and the categorization of coded data.

### 2.6. Ethical Consideration

Ethical approval was obtained from [blind for peer review] University’s institutional review boards. Prior to data collection, consent/assent was obtained from each participant after explaining risks and benefits, confidentiality, and options for withdrawal.

## 3. Findings

Theme 1. From Textbook to Providing Humanistic Care in Low-Resourced Work Contexts

There is a gap between the knowledge on humanistic care gained from school curricula and implementing it in practice, particularly in the settings of general hospitals with low nurse-to-patient ratios and high intensity of work. Nursing students are not prepared to implement humanistic care in a task-oriented and stressful context.


*“Too Busy to Provide Humanistic care”?*


Many students questioned themselves on how to balance completing nursing tasks and providing humanistic care, which was not taught in class. Some participants noticed when the nurse-to-patient ratio is good, nurses have more time to integrate a humanistic spirit into their care,


*“Patients in the VIP unit tend to more humanistic care from health care professionals because nurses and their co-workers have more time to attend to their needs and communicate with them compared to normal units that are always full and noisy. The charges are higher of course.”*
(P19)

Some students described potential strategies learned from their observation and experience, including knowing your patients well, knowing how to organize your work by priority, and, most importantly, empowering patients. They believe that nurses who are good at their work know their patients well, are able to anticipate patients’ needs, and be prepared in advance. In this way, nurses can prioritize and organize their work in a more efficient manner. Nurses can empower some patients and family members to do appropriate self-care work so that they can have more time to attend to patients’ higher hierarchy of needs. Many students agreed that humanistic care does not simply equal being gentle and doing everything for patients. In real clinical settings, nurses cannot just sit there and talk to patients or help them with everything. They need to empower patients and their families to do what they are capable of by giving clear instructions. It is not bossing patients around or trying to do less work. It is empowering them and supporting them to do self-care:


*“When you learn humanistic care from textbook in school, you feel that you need to be very gentle and supportive, do everything for patients, and communicate with them, but in fact, most importantly, you need to be able to empower patients. This is not what I learned from class, but it is extremely important to do so in a busy unit.”*
(P16)


*An Inseparable Part of Daily Nursing Practice and Professionalism*


Some participants automatically separated humanistic care from nursing procedures and daily work and considered it as extra work that nurses have to do. Some students admitted that they focused exclusively on practicing and improving their abilities in performing nursing procedures during their practicum, and one participant believed that: “being skilled in nursing procedures and practices itself is considered as humanistic care.” As some of them had more experience and observations in clinical settings, they later realized that integrating humanistic care into daily nursing practice is a prerequisite for implementing humanistic care in real clinical settings. Nurses do not regard implementing humanistic care as extra work but internalize it and embed it in day-to-day work:

*“as a nurse, you have to figure out how to internalize the value of humanistic care and integrate it into your professional knowledge and nursing practices.”* (P03) *“Nurses who have done well in humanistic care in my observation are those who do not regard humanistic care as extra work to be completed on the daily list, but embed humanistic care in daily practice until it become a habit of their own.”*(P05)

More specifically, humanistic care can be seen throughout the whole nursing process: from admission to discharge, in assessments, in verbal and non-verbal communications, in the decision-making process, and in every nursing procedure. Many participants regard humanistic care as holistic and that as nurses, we must see patients as whole persons who deserve full respect:


*“we need to put ourselves in patients’ shoes, see them as whole persons who need holistic care to maintain health and wellbeing.”*
(P32)


*“Person-Centered or Task-oriented”?*


Respecting the personhood of each patient and integrating person-centeredness in daily care is a key element of humanistic care in the eyes of nursing students. They believe that every patient and their family deserves humanistic care and nurses should “act in the interest of patients.” They all learned humanistic care from textbooks in school, but when they reflected on their observation and experiences in clinical settings, they realized that nursing students could not simply apply what they learned from school to practice. Instead, they need to get to know each patient and tailor humanistic care to the personalities and needs of each patient:

*“the key is to tailor it to the needs and characteristics of each patient. You may think that regularly checking in and talking with patients would be considered a way of providing humanistic care but it does not work for everyone. If you just apply what you learned or considered as humanistic care for patients without taking their needs into consideration, good intentions could lead to negative results.”* (P11) *“Humanistic care needs to be flexible and patient-oriented rather than textbook-oriented.”*(P10)

One student shadowed a nurse in the cardiology unit, and she was impressed by how much the nurse knew and cared about her patients. The nurse did not just check the task list but paid special attention to the needs of her patients. The nurse identified patients who were more vulnerable to having anxiety or depressive symptoms based on her understanding of their conditions and observed their changes closely to be able to integrate humanistic care into her everyday work, helping patients maintain psychological well-being.


*Challenges in Evaluating Humanistic Care*


Participants identified challenges in evaluating humanistic care in clinical settings. There are two primary means of evaluation, including a self-administered checklist developed by hospitals for nurses and a satisfaction survey for patients and families. The checklists vary in different hospitals and are not developed following psychometric principles to ensure validity and reliability. Items appearing on the checklist include frequency and duration of communication with patients, frequency of checking in with patients, and some other items that can be quantified and recorded as a proxy of the implementation of humanistic care considered by the hospitals. However, ambiguity in how to evaluate humanistic care remains, and the core concepts and mentality of humanistic care are missing from the checklist.

Some managers and head nurses rely heavily on patients’ general satisfaction survey results to evaluate the quality of humanistic care. Students observed that multiple factors can work together and impact the level of patients’ satisfaction so that when managers and head nurses use it as the only indicator to evaluate humanistic care, there can be bias:


*“the satisfaction results can be influenced by many other factors. Patients and family may forget to fill out the printed survey or just checked the items without reading it carefully at discharge. And sometimes, nurses would read the items to patients and check the items for them because some patients cannot read, however, these patients can feel pressured that they have to say something good.”*
(P48)

Theme 2. Building Trust between Patients and Healthcare Professionals through Providing Humanistic Care

Participants shared with interviewers about their reflections and thoughts on some potential ethical issues in clinical settings. A few students witnessed conflicts/violent incidents perpetrated by patients due to multiple reasons, such as misunderstandings, denying family members’ death, and so forth. How to “protect ourselves” as health care professionals and to provide quality care for patients were consistently mentioned by our participants. They believe, ideally, humanistic care should contribute to the relationship between health care professionals and patients and avoid conflicts if implemented appropriately.


*“I witnessed a physical conflict in the unit. One patient was very sick when sent to the unit. We tried our best, but he died. One of the nurses informed the family members and tried to comfort them, but family members refused to accept it and they got violent. I was terrified and was thinking that as nurses, we need to protect ourselves even when we are trying to communicate and provide humanistic care for patients or family members. It is like walking on ice.”*
(P01)

Humanistic care does not simply mean meeting patients’ every single need. It requires nurses to collaborate with their co-workers, assess the changes of patients’ needs and challenges accurately, and make well-informed decisions. One student commented that providing humanistic care did not simply mean nurses had to meet every requirement from patients. Instead, nurses need to observe, assess, communicate with patients and their co-workers, obtain adequate information, facilitate knowledge-sharing in the work context and make shared decisions.


*“When I first saw patients or family members crying, I got emotional and would try my best to comfort them. However, later, I realize that I need to be professional and I cannot promise things that won’t happen or are inappropriate just to make patients or family members feel better. It is not that I become cold-hearted, it is how you provide humanistic care.”*
(P61)

One student shared her reflections on communicating with patients and establishing trust, stating that nurses cannot *judge* patients’ beliefs or values. For example, when a patient with advanced cancer believes that a certain herb tea would cure his/her disease, even though nurses know it is impossible, we do not judge them or tell them that it will not work, as long as it does not do harm or contradict with medicines that they are using.

Theme 3. Implications for Humanistic Nursing Care Education from Students’ Perspectives

We asked participants to reflect on their experiences in school and in clinical practicum and provide some advice for the improvement of humanistic care education focusing on patients for undergraduate students. They consistently mentioned the importance of helping students understand and internalize the essence of humanistic care and preparing them to be able to acquire knowledge in clinical practicum through observation, critical thinking, and phenomenological reflections. When some students first started their practicum, they believed that nurses’ most important job was to provide professional and quality care rather than being humanistic, so they focused on practicing nursing techniques and procedures. Later, they realized that humanistic care is a core part of quality care and all their role model nurses in clinical settings are good at providing humanistic care for patients and families.


*“I wish I had known it better. In the complex context of clinical settings, I was overwhelmed and I wish I had practiced taking an leadership role in learning, to actively learn rather than just waiting to be taught by others.”*
(P33)

Some identified the gap between what they learned in school and the competency needed to implement humanistic care in clinical practices where nurses and nursing students are faced with contextual barriers and challenges such as high intensity of work, inadequate staff, and a task-oriented culture. They also suggested that nursing educators in undergraduate programs collaborate closely with clinical nurses at an earlier stage to provide positive guidance for students to integrate humanistic care into their learning and practicum through self-introspection, reflective thinking, group discussion, and leadership. Some students felt that they were living in a bubble and had idealistic thoughts about what humanistic care is. A few students were pessimistic about the nursing profession and how unrealistic it is to provide humanistic care in clinical settings. Other students pointed out that:


*“Being pessimistic does not help. We need to be actively guided to understand the difficulty and the opportunities of doing things better as the next generation of nurses. We should also talk about risks and how to protect ourselves if it happens. I want to be better prepared and be a good nurse.”*
(P87)

There needs to be a balance between the scientific and technical aspects of nursing and the significant yet often-neglected person-centered and humanistic aspects, which align well with the core elements of nursing science and the profession. Many participants agree that this is a long-term and continuous effort and exploration amongst nursing educators and students. Many students mentioned that they learned of humanistic care in school. The care model was mentioned repeatedly and embedded in some core curricula. However, what humanistic care is and how to integrate it into daily practice were vague to them:


*“I hope future students can learn as early as possible how to provide humanistic care in real world by having interactions with clinical nurses and patients through integrated practicum, case studies, group discussions and critical reflections. I hope we can have the chance to take a lead and think about how to do it when we have fresh eyes before we are told how to do it.”*
(P16)

Some students suggested that nursing schools collaborate with teaching hospitals and develop a strategic plan to facilitate students’ proactive role and leadership behaviors in providing humanistic care in clinical settings during their internship. They believe that as students, they have time, patience and curiosity, which are their strengths, and need to be well used by the teaching hospitals to make them leaders in providing humanistic care for patients:


*“we are not charge nurse, we don’t have tasks to complete. We are here to learn and practice. I actually think the hospitals should let us be an important part of providing humanistic care and communicate with patients, but of course, we need some guidance and supervision from the charge nurses. I believe school and our teaching hospitals can empower students like us to play an active part in delivering quality care.”*
(P92)

## 4. Discussion

The current qualitative study adopted a participatory approach to facilitate the direct engagement of key stakeholders in the study design, data collection, and interpretation. Participants actively shared their experiences and critical reflections on humanistic nursing care education in their transitions from school to clinical practicum in a low-resourced context. This study is among the few articles that provide critical reflections on learning and delivering humanistic care and humanistic education based on students’ experiences and perspectives and has international relevance in low- and middle-income countries within a low-resourced care context.

Some students identified a gap between what they learned in school and the competency needed to implement humanistic care in clinical practices where nurses and nursing students are faced with contextual barriers and challenges such as high intensity of work, inadequate staff, and a task-oriented culture. Additionally, if students wanted to learn about providing humanistic care in their practicum, they had to do it through observations and asking ward nurses questions without positive guidance. Top schools of nursing in China have included more humanistic spirits in the curricula and emphasized the importance of humanistic nursing care in recent years. As a result, most students have some knowledge of humanistic nursing care. However, when students are placed in a stressful clinical context directly from school, they become overwhelmed and can be confused by the gap between what is happening in their practicum and what they learned from school, particularly when there is no positive guidance from their clinical instructors. It is significant to systematically and sustainably integrate the spirit and values of humanistic care into the development plan and teaching activities in clinical learning environments, to help students understand and internalize the essence of humanistic care and prepare them to be able to acquire knowledge in the clinical practicum.

Our findings suggest that nursing educators in undergraduate programs collaborate closely with clinical instructors who provide direct supervision and instruction to nursing students during their clinical practicum at an earlier stage to provide positive guidance for students to integrate humanistic care into their learning and practicum through self-introspection, reflective thinking, group discussion, and leaderships. Clinical instructors and school faculty members can collaborate in developing curricula and clinical practicum plans and creating opportunities for students to apply their knowledge. Students can benefit from being engaged in early interactions or in-depth dialogue with patients and families to have practical experiences and interest in enhancing their competency. Person-centered education methods, the concept-based method, and interactive humanistic methods were suggested by other researchers to help students integrate their experiences and internalize a humanistic value system through self-introspection, group discussion, and reflective thinking [5,8,10,11,21]. However, the current core curricula are still fact- and skill-oriented. There needs to be a balance between the scientific and technical aspects of nursing and the significant, yet often neglected, person-centered and humanistic aspects, which align well with the core elements of nursing science and the profession. Pedagogical strategies currently in use may need to be modified to accommodate humanistic nursing care in low-resourced and challenging healthcare environments. The evaluation of students’ competency may also be adapted to move beyond techniques and skills and focus more on their capabilities for flexibility and creativity in nursing care across settings, problem-solving, critical and abstract thinking, collaboration, and evidence use.

Students who have acquired and understood the essence of humanistic care realized the importance of integrating humanistic care into daily nursing practice, respecting the personhood of each patient, and embedding person-centeredness by tailoring humanistic care to the needs of each patient through their observations and reflective thinking in their practicum. However, they felt confused and concerned about some potential ethical issues in clinical settings such as protecting themselves from potential conflict. They also questioned themselves about how to spare time from completing routine nursing tasks to provide more humanistic care for patients and family. When we move beyond education and training, we can see that the high intensity of nursing work, the inadequate nursing staffing, and the medically-dominated and task-oriented culture are contextual barriers to providing and learning humanistic nursing care in clinical settings. Nurses are in a dilemma between providing quality and person-centered care for patients and facilitating students’ learning while competing their daily tasks. A lack of valid tools to evaluate the quality of humanistic care adds to the challenge of motivating nurses to provide humanistic care. Systematic reforms are needed to make clinical settings more humanistic care-friendly for nurses and nursing students [9,13].

## 5. Conclusions

There is a gap between what undergraduate nursing students learned in school and the competency needed to implement humanistic care in clinical practices when faced with contextual barriers and challenges such as high intensity of work, inadequate staff, and a task-oriented culture. It is significant for nurse educators/instructors working in clinical learning environments to systematically integrate the spirit and values into the teaching activities to help students understand and internalize the essence of humanistic care and prepare them to be able to acquire knowledge in clinical practicums. At the same time, pedagogical strategies currently in use may need to be modified to accommodate humanistic nursing care in low-resourced and challenging health care environments. The evaluation of students’ competency should also be adapted to move beyond techniques and skills and focus more on their capability of flexibility and creativity in nursing care across settings. Systematic reforms are needed to make clinical settings more humanistic care-friendly for nurses and nursing students and pave the way for the reforms in education and professional development. The current study provided critical reflections on learning and delivering humanistic care and humanistic education based on students’ experiences and perspectives, and has international relevance for providing humanistic nursing care in low-resourced care contexts, where financial resources and nursing staffing are low.

## Figures and Tables

**Table 1 ijerph-19-12656-t001:** Characteristics of Participants (*N* = 120).

Characteristics	*N* (%)
Age (interquartile Range)	20 (20–22)
Sex	
Male	25 (21)
Female	95 (79)
Only Child or not	
Yes	98 (82)
No	22 (18)

## Data Availability

The data presented in this study are available on request from the corresponding author.

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
