# Peer review of "Humanistic Nursing Care for Patients in Low-Resourced Clinical Settings from Students’ Perspectives: A Participatory Qualitative Study"

_ijerph, 2022, doi:10.3390/ijerph191912656_

Round 1

Reviewer 1 Report

Thank you for this opportunity to get to know your work. Congratulations, you have produced new scientific knowledge for nursing discipline, and applied research method that is rarely used in nursing science - good job! I have made some remarks about your manuscript. I do hope that you will find my following comments useful when improving the quality of your manuscript.

Title: I ask the authors to re-think the title of this manuscript. First and foremost, I would like to point out that, the tittle should also include the following central aspects of this study: "learning and delivering humanistic nursing care”, because if you compare the research aim (rows 63-68) and introduction, you can find those aspects from there. Also, the “critical reflections on humanistic education” is included in your research aim. Therefore, I would also add it up in the tittle. At least I guess that is your research aim which is stated in rows 63-68? Lastly, there is “perspective” mentioned in the tittle, but authors are also using “experiences”, so what is the relation between perception and experience? Another example of this can be found in rows 115-116. Please consider which one will be appropriate for this topic, and check that you will use systematically the same expression throughout the manuscript, for example “experiences”.

Abstract: The authors have defined following keywords in abstract: Humanistic Care; Patients; Low-resourced environment; Nursing Students; Ethical Considerations. However, the keyword “Ethical Considerations” is being used only once in the abstract. Hence, I would suggest that the authors would re-consider using that keyword. In addition, I would like to point out that there is no keyword(s) regarding research method of this study – please add up. Also, I would suggest that “clinical practicum” could be used as a key word. If possible, please define “humanistic care” and “low-resourced healthcare environment” in the abstract. Definition could be done with examples. Should your research aim also include “the critical reflections of humanistic education”? There should be a number of participants included in the abstract, and a mention that there were also educators as participants. Also, please include the time when data was collected.

Introduction: The topic involves an important perspective for nursing discipline, and it has value for development of nursing education, as well as for student supervision in hospitals. However, I think that the authors could strengthen the introduction by describing in more detail why this scientific knowledge regarding students´ perspectives was needed. I mean you could point out the knowledge gap in previous literature in more detail. Now the introduction focuses mainly on humanistic care and its implementation to nursing schools. Authors have defined humanistic care in rows 26-28 which is good. However, I was wondering how does this humanistic nursing care differ for "regular nursing"? I mean could you please elaborate, what is the relation of humanistic nursing care and nursing care? In row 47 you mention about the humanistic care model, could you please describe this model briefly. Lastly, this goes both the introduction and the conclusion: could you please point out what is the international relevance of this study?

Am of the study (rows 63-68??): See my previous comments earlier related to the research aim. I argue that your research aim should be expressed clearly, for example in the form of “The aim of this study was to explore…”. I would suggest that you would add up here research questions. Maybe in total of three research questions would be needed (experiences of learning and delivering…. and critical reflections on…). I think that it would benefit this manuscript.  

Materials and methods: The study has been planned and implemented by applying scientific research methods and selection of those methods is justified. Still, I would like to address some methodological issues. Number of participants should be clearly expressed at each phase of this participatory study, in a form of (n=xx). Also, please state clearly if those are students, educators or others, e.g. (students n=xx), (educators n=xx). At the beginning of section 2.4., please correct the method of analysis. I do not consider “conventional” as an appropriate term for analysis. What is non-conventional then? Also, please specify the approach used in the qualitative content analysis.

Rigor: I would like to ask the authors to critically review the rigor of this study. Now the rigor of this study is reviewed superficially. In addition, the authors could use references to support their argumentation regarding the trustworthiness of this qualitative study. For instance, it would be much needed that the authors would use checklists developed for these purposes, such as COREQ (COnsolidated criteria for REporting Qualitative research) Checklist). Lastly, the rigor should be reviewed in terms of this participatory qualitative method.

Findings: Could you please re-think the naming of the theme 2 “Ethical consideration”. That should me more descriptive tittle. What does this theme reflect? What kind of ethical considerations? From whose perspective and so on?

Discussion: Please indicate, at the end of your first paragraph, what is the novelty value of your research results compared to what is previously known about this topic. When doing so, please include all your point of views: learning and delivering humanistic care, critical reflections of humanistic education, and low-resourced clinical settings. In row 353, please define who is “clinical instructor”. Since nursing education is carried out differently across the world, also the concepts regarding nursing education are being used variously. Therefore, it is essential to define those concepts to make readers understand your argumentation, and so that they can use this information correctly.

Conclusions: The conclusions are justified based on the results. However, it would be a great value for this manuscript if the authors could provide some recommendations nurse educators and nurse preceptors/mentors working in clinical learning environments. In row 393, please indicate whose teaching activities are you referring on? Preceptor´s / mentor´s / educator´s? Lastly, state what is international relevance of this paper?

To sum up, the structure of the manuscript is consistent. Clarity and readability are good. Although, this manuscript could benefit (English) language editing and proof-reading, because there are space errors throughout the text (e.g., in rows 33, 258, 215, 281, 117 and so on).

I wish you all the best with revising your manuscript!

Author Response

Thank you for this opportunity to get to know your work. Congratulations, you have produced new scientific knowledge for nursing discipline, and applied research method that is rarely used in nursing science - good job! I have made some remarks about your manuscript. I do hope that you will find my following comments useful when improving the quality of your manuscript.

Title: I ask the authors to re-think the title of this manuscript. First and foremost, I would like to point out that, the tittle should also include the following central aspects of this study: "learning and delivering humanistic nursing care”, because if you compare the research aim (rows 63-68) and introduction, you can find those aspects from there. Also, the “critical reflections on humanistic education” is included in your research aim. Therefore, I would also add it up in the tittle. At least I guess that is your research aim which is stated in rows 63-68? Lastly, there is “perspective” mentioned in the tittle, but authors are also using “experiences”, so what is the relation between perception and experience? Another example of this can be found in rows 115-116. Please consider which one will be appropriate for this topic, and check that you will use systematically the same expression throughout the manuscript, for example “experiences”.

Response: Thank you for the comment. I revised the title to Learning humanistic nursing care for patients in low-resourced clinical settings from students’ experiences and reflections: a participatory qualitative study. I also made sure that the same expression was used consistently throughout the manuscript.

Abstract: The authors have defined following keywords in abstract: Humanistic Care; Patients; Low-resourced environment; Nursing Students; Ethical Considerations. However, the keyword “Ethical Considerations” is being used only once in the abstract. Hence, I would suggest that the authors would re-consider using that keyword. In addition, I would like to point out that there is no keyword(s) regarding research method of this study – please add up. Also, I would suggest that “clinical practicum” could be used as a key word. If possible, please define “humanistic care” and “low-resourced healthcare environment” in the abstract. Definition could be done with examples. Should your research aim also include “the critical reflections of humanistic education”? There should be a number of participants included in the abstract, and a mention that there were also educators as participants. Also, please include the time when data was collected.

Responses: Thank you for the comments. I deleted ethical consideration from the keywords list. I also added participatory qualitative study and clinical practicum to the keywords list. We added the information that. “A total of 120 eligible undergraduate students were included in the study.” Faculty members participated in the data analyses but not as study participants during data collection. I also added data collection time at the end of methods of abstract. We defined “humanistic care” as “a core element of quality care for patients/family characterized by empathy and holistic care and “low-resourced healthcare environment” as “with seriously low nursing staffing level” in the abstract.

Introduction: The topic involves an important perspective for nursing discipline, and it has value for development of nursing education, as well as for student supervision in hospitals. However, I think that the authors could strengthen the introduction by describing in more detail why this scientific knowledge regarding students´ perspectives was needed. I mean you could point out the knowledge gap in previous literature in more detail. Now the introduction focuses mainly on humanistic care and its implementation to nursing schools. Authors have defined humanistic care in rows 26-28 which is good. However, I was wondering how does this humanistic nursing care differ for "regular nursing"? I mean could you please elaborate, what is the relation of humanistic nursing care and nursing care? In row 47 you mention about the humanistic care model, could you please describe this model briefly. Lastly, this goes both the introduction and the conclusion: could you please point out what is the international relevance of this study?

Responses: Thank you for the comment. For the importance of having students’ perspectives, we added the information as follows, “It is of utmost importance to understand from students’ perspectives so that we can develop a student-centered nursing education program with more applicability that can better prepare our students to value humanistic spirits and provide humanistic care in their work.” We also added the information regarding humanistic care as follows, “Compared to regular nursing care, humanistic care shifted from a task-oriented care model to a person-centered or relationship-centered model.” In terms of humanistic care model, we added the information as follows, “…The humanistic care model was only mentioned in the current nursing curricula, however, it is abstract and not incorporated in the key learning contents or grounded in clinical teaching.”  We added international relevance in both introduction and conclusion.

Am of the study (rows 63-68??): See my previous comments earlier related to the research aim. I argue that your research aim should be expressed clearly, for example in the form of “The aim of this study was to explore…”. I would suggest that you would add up here research questions. Maybe in total of three research questions would be needed (experiences of learning and delivering…. and critical reflections on…). I think that it would benefit this manuscript.  

Responses: Thank you for the comments.  I added the study aim as, “The aim of the study was to explore nursing students’ experiences of learning and delivering humanistic care during their clinical practicum and the critical reflections on humanistic care education.

Materials and methods: The study has been planned and implemented by applying scientific research methods and selection of those methods is justified. Still, I would like to address some methodological issues. Number of participants should be clearly expressed at each phase of this participatory study, in a form of (n=xx). Also, please state clearly if those are students, educators or others, e.g. (students n=xx), (educators n=xx). At the beginning of section 2.4., please correct the method of analysis. I do not consider “conventional” as an appropriate term for analysis. What is non-conventional then? Also, please specify the approach used in the qualitative content analysis.

Responses: Thank you for the comments. The number of participants were added in the form of n=120. Only students were included as participants in the data collection.  I cited the methodological article from Hsieh, H. F., & Shannon, S. E. (2005). Three approaches to qualitative content analysis. Qualitative health research. https://doi.org/10.1177/1049732305276687, which described three distinct content analysis approaches to descriptive qualitative study: conventional, directed, or summative analyses. All three approaches are used to interpret meaning from the content of text data and, hence, adhere to the naturalistic paradigm. I chose using conventional content analysis because in conventional content analysis, coding categories are derived directly from the text data.

Rigor: I would like to ask the authors to critically review the rigor of this study. Now the rigor of this study is reviewed superficially. In addition, the authors could use references to support their argumentation regarding the trustworthiness of this qualitative study. For instance, it would be much needed that the authors would use checklists developed for these purposes, such as COREQ (COnsolidated criteria for REporting Qualitative research) Checklist). Lastly, the rigor should be reviewed in terms of this participatory qualitative method.

Responses: Thank you for the comment. We cited the articles from Sandelowski and Lincoln and Guba that discussed about qualitative study rigor. We also used COREQ to check before submission. We revised the rigor as follows by also taking participatory qualitative approach into account, “Multiple strategies were used to establish qualitative rigor. For credibility (confidence in the truth of the findings), we used multiple data sources including students’ journal of participatory observation and reflections, and experiences shared in the interviews. We adopted participatory qualitative approach so that students had prolonged engagement and persistent observation that can provides scope and depth to the study.  To further ensure confirmability (a degree of neutrality), we held frequent meetings of the coding team to discuss the use of codes, the coding, and refined definitions needed. We also engaged other non-researcher team members so that they could review and challenge our interpretations. For transferability (applicability), students were encouraged to keep detailed account of field experiences so that the researcher can understand the context more explicitly. For dependability (findings are consistent and could be repeated), we kept summaries and memos of all the discussion among members of the coding team to keep track of the interpretation. The refinement of the codes and themes were discussed by the whole team, which facilitated joint decision making the coding schemes and categorization of coded data.

Findings: Could you please re-think the naming of the theme 2 “Ethical consideration”. That should me more descriptive tittle. What does this theme reflect? What kind of ethical considerations? From whose perspective and so on?

Response: Thank you for the comment. I revised the second theme to “building trust between patients and health care professionals through delivering humanistic care”.

Discussion: Please indicate, at the end of your first paragraph, what is the novelty value of your research results compared to what is previously known about this topic. When doing so, please include all your point of views: learning and delivering humanistic care, critical reflections of humanistic education, and low-resourced clinical settings. In row 353, please define who is “clinical instructor”. Since nursing education is carried out differently across the world, also the concepts regarding nursing education are being used variously. Therefore, it is essential to define those concepts to make readers understand your argumentation, and so that they can use this information correctly.

Responses: Thank you for your comment. I added the information at the end of the first paragraph as follows, “It is among the few articles that provided critical reflections on learning and delivering humanistic care and humanistic education based on students’ experiences and perspectives and has international relevance in low- and middle-income countries with low-resourced care context.” 

Thank you for your comment. We agree that it is essential to define key concepts so that readers can use the evidence correctly. Clinical instructors are defined as “clinical nurses taking the role as clinical educators who provides direct supervision and instruction to nursing students during their clinical practicum”.

Conclusions: The conclusions are justified based on the results. However, it would be a great value for this manuscript if the authors could provide some recommendations nurse educators and nurse preceptors/mentors working in clinical learning environments. In row 393, please indicate whose teaching activities are you referring on? Preceptor´s / mentor´s / educator´s? Lastly, state what is international relevance of this paper?

Responses: Thank you for your comments. I revised the conclusions as follows, “It is significant for nurse educators/instructors working in clinical learning environment to systematically integrate the spirit and values into the teaching activities and supervision to help students understand and internalize the essence of humanistic care and prepare them to be able to acquire knowledge in clinical practicum.” In terms of international relevance, “The current study provided critical reflections on learning and delivering humanistic care and humanistic education based on students’ experiences and perspectives and has international relevance for low- and middle-income countries in terms of providing humanistic nursing care in low-resourced care context, where the financial resource and nursing staffing is low.” 

To sum up, the structure of the manuscript is consistent. Clarity and readability are good. Although, this manuscript could benefit (English) language editing and proof-reading, because there are space errors throughout the text (e.g., in rows 33, 258, 215, 281, 117 and so on).

Response: Thank you for the comments. I went through the manuscript and corrected the space errors.

Reviewer 2 Report

Dear authors!

Thank you for the opportunity of reviewing this manuscript about the humanistic nursing care for patients. I think that humanistic nursing care is a core of quality care, and more research should cover this important topic. Therefore, I welcome the efforts of the author in carrying out the research. This study is interesting, and I am glad to read it. I wish my comments will help to improve the manuscript.

Introduction:

Please describe in more detail what humanistic nursing care really is.

Line 30-32 please insert the reference for statement that humanistic nursing care can improve safety and quality of care.

 Methods:

 Please provide additional information on how to recruit participants. Please consider adding such information as a) the recruitment date range (month and year), b) a description of any inclusion/exclusion criteria that were applied to participant recruitment. You mentioned that participants are from a top School of Nursing. It is unclear what this means and how you chose them. Table 1 that you mentioned  (line 95) cannot be found.

Please edit the manuscript using backspace between some words.

Author Response

Dear authors!

Thank you for the opportunity of reviewing this manuscript about the humanistic nursing care for patients. I think that humanistic nursing care is a core of quality care, and more research should cover this important topic. Therefore, I welcome the efforts of the author in carrying out the research. This study is interesting, and I am glad to read it. I wish my comments will help to improve the manuscript.

Introduction:

Please describe in more detail what humanistic nursing care really is.

Responses: Thank you for your comments. We added more description in the introduction regarding humanistic nursing care and compared it with regular nursing care. See the highlighted text in introduction.

Line 30-32 please insert the reference for statement that humanistic nursing care can improve safety and quality of care.

Responses: Thank you for your comments. We added the citations for the statement.

 Methods:

 Please provide additional information on how to recruit participants. Please consider adding such information as a) the recruitment date range (month and year), b) a description of any inclusion/exclusion criteria that were applied to participant recruitment. You mentioned that participants are from a top School of Nursing. It is unclear what this means and how you chose them. Table 1 that you mentioned (line 95) cannot be found.

Responses: Thank you for your comment. We added the information as follows, “All the eligible students who started their clinical practicum in 2020 was included in the study except one who dropped out of the practicum due to personal reasons.”  The reasons that we chose the school of nursing were that it emphasized humanistic care in its program description and curricula. It ranked in the first place of all the schools of nursing in China. “We included all the third-year undergraduate nursing students who started their clinical practicum in 2020, from a School of Nursing in China that emphasizes humanistic care in its program descriptions (See Table 1). Data collection started from March 2020 to June 2021.”  Table 1 was in a separate document. I included the table below in the response letter.

Please edit the manuscript using backspace between some words.

Responses: Thank you for your comments. We edited the space problem between some words throughout the manuscript.
